# The Role of Toll-like Receptor-4 in Macrophage Imbalance in Lethal COVID-19 Lung Disease, and Its Correlation with Galectin-3

**DOI:** 10.3390/ijms241713259

**Published:** 2023-08-26

**Authors:** Maria Carmela Pedicillo, Ilenia Sara De Stefano, Rosanna Zamparese, Raffaele Barile, Mario Meccariello, Alessio Agostinone, Giuliana Villani, Tommaso Colangelo, Gaetano Serviddio, Tommaso Cassano, Andrea Ronchi, Renato Franco, Paola Pannone, Federica Zito Marino, Francesco Miele, Maurizio Municinò, Giuseppe Pannone

**Affiliations:** 1Department of Clinical and Experimental Medicine, University of Foggia, Viale L.Pinto 1, 71122 Foggia, Italy; mariacamela.pedicillo@unifg.it (M.C.P.); ileniadestefano@hotmail.it (I.S.D.S.); alessio.agostinone@gmail.com (A.A.); 2Legal Medicine Unit, Ascoli Piceno Hospital C-G. Mazzoni, Viale Degli Iris 13, 63100 Ascoli Piceno, Italy; rosanna.zamparese@sanita.marche.it; 3Department of Medical and Surgical Sciences, University of Foggia, Viale L.Pinto 1, 71122 Foggia, Italy; raffaele_barile.552733@unifg.it (R.B.); mario_meccariello.552823@unifg.it (M.M.); tommaso.colangelo@unifg.it (T.C.); gaetano.serviddio@unifg.it (G.S.); tommaso.cassano@unifg.it (T.C.); 4Policlinico Riuniti, University-Hospital, Viale L.Pinto 1, 71122 Foggia, Italy; giuliana-91@hotmail.it; 5Cancer Cell Signalling Unit, Institute for Stem-Cell Biology, Regenerative Medicine and Innovative Therapies (ISBReMIT), IRCCS Fondazione Casa Sollievo della Sofferenza, Viale Cappuccini sc.c., San Giovanni Rotondo, 71013 Foggia, Italy; 6Pathology Unit, Department of Mental and Physical Health and Preventive Medicine, University of Campania “L Vanvitelli”, via Luciano Armanni, 80138 Naples, Italy; andrea.ronchi@unicampania.it (A.R.); renato.franco@unicampania.it (R.F.); federicazito.marino@libero.it (F.Z.M.); 7Federico II, Department of Clinical Medicine and Surgery, School of medicine and Surgery, University of Naples, via Sergio Pasini, 80131 Naples, Italy; pao.pannone@studenti.unina.it; 8Department of Surgery, University of Campania “L Vanvitelli”, 80138 Naples, Italy; dott.miele@gmail.co; 9Forensic Medicine Unit, “S. Giuliano” Hospital, via Giambattista Basile, 80014 Giugliano in Campania, Italy; maurizio.municino@aslnapoli2nord.it

**Keywords:** TLRs, TLR-4, SARS-CoV-2, lethal COVID-19, lung disease, macrophage shift, ARDS, CD68, galectin-3, CD163, M1 macrophages, M2 macrophages

## Abstract

To the current data, there have been 6,955,141 COVID-19-related deaths worldwide, reported to WHO. Toll-like receptors (TLRs) implicated in bacterial and virus sensing could be a crosstalk between activation of persistent innate-immune inflammation, and macrophage’s sub-population alterations, implicated in cytokine storm, macrophage over-activation syndrome, unresolved Acute Respiratory Disease Syndrome (ARDS), and death. The aim of this study is to demonstrate the association between Toll-like-receptor-4 (TLR-4)-induced inflammation and macrophage imbalance in the lung inflammatory infiltrate of lethal COVID-19 disease. Twenty-five cases of autopsy lung tissues were studied by digital pathology-based immunohistochemistry to evaluate expression levels of TLR-4 (CD 284), pan-macrophage marker CD68 (clone KP1), sub-population marker related to alveolar macrophage Galectin-3 (GAL-3) (clone 9C4), and myeloid derived CD163 (clone MRQ-26), respectively. SARS-CoV-2 viral persistence has been evaluated by in situ hybridation (ISH) method. This study showed TLR-4 up-regulation in a subgroup of patients, increased macrophage infiltration in both Spike-1^(+)^ and Spike-1^(−)^ lungs (*p* < 0.0001), and a macrophage shift with important down-regulation of GAL-3^(+)^ alveolar macrophages associated with Spike-1 persistence (*p* < 0.05), in favor of CD163^(+)^ myeloid derived monocyte-macrophages. Data show that TLR-4 expression induces a persistent activation of the inflammation, with inefficient resolution, and pathological macrophage shift, thus explaining one of the mechanisms of lethal COVID-19.

## 1. Introduction

The innate immune response is a key issue to act as the first line of defense against many viral infections [1]. The host immune response to SARS-CoV-2 infection is critical in determining the severity of COVID-19 [2].

It is assumed that cytokine storms, which are the consequence of hyperinflammation driven by innate immunity, play an important role in the pathogenesis of severe SARS-CoV-2 [3], however, the underlying mechanisms of the altered pathological inflammation in COVID-19 remain largely unknown.

The key innate immune sensing receptors are germ line-encoded pattern-recognition receptors (PRRs), which mediate the initial sensing of infection by recognition of PAMPs (pathogen-associated molecular patterns), upon microbial invasion of the host [4].

PRRs belong to different families, including Toll-like-receptors (TLRs).

TLRs are the most studied in the families of PRRs, and Toll-like receptor-4 (TLR-4) is one of the most studied. TLRs specifically recognizes bacterial lipopolysaccharide (LPS), and its activation mainly leads to the synthesis of pro-inflammatory cytokines and chemokines [5,6].

TLRs are the important mediators of inflammatory pathways in the gut, playing a major role in mediating the immune responses towards a wide variety of pathogen-derived ligands and link adaptive immunity with the innate immunity [7]. TLRs are directly involved in the regulation of inflammatory reactions and activation of the innate or adaptive immune responses for the elimination of infectious pathogens [8,9].

TLR-4 is a protein that in humans is encoded by the TLR-4 gene. TLR-4 is a transmembrane protein, member of the TLRs family, which belongs to the PRR family. TLR-4 activation leads to an intracellular signaling pathway NF-KB and inflammatory cytokine production, which is responsible for activating the innate immune system [10].

TLR-4 expressing cells are myeloid (erythrocytes, granulocytes, macrophages) rather than lymphoid (T-cells, B-cells, NK cells). Most myeloid cells also express high levels of CD14, which facilitates activation of TLR-4 by LPS.

It is well known for recognizing LPS, a component present in many Gram-negative bacteria (e.g., *Neisseria* spp.), and selected Gram-positive bacteria. TLR-4 ligands also include several viral proteins, polysaccharide, and a variety of endogenous proteins such as low-density lipoprotein, beta-defensins, and heat shock protein [11].

A recent study reported that alveolar macrophages activated by COVID-19 through TLR signaling produced interleukin-1 (IL-1), which further stimulates mast cells to produce interleukin-6 (IL-6) [12]. In addition, an association among mast cell, eosinophil activation, and COVID-19 inflammation was reported [13]. A significantly elevated expression of IL-6 and tumor necrosis factor-α (TNF-α) was found to be associated with TLR expression in obese individuals but not in the controls [14], a well-known cofactor associated with increased risk of mortality in COVID-19 disease.

In silico studies indicated that cell surface TLRs, mainly TLR-4, are most likely to be involved in sensing molecular patterns, including SARS-CoV-2 S protein, to induce inflammatory responses [15,16]. It was been reported that the SARS-CoV-2 S protein S1 subunit can induce pro-inflammatory cytokines through TLR-4 signalling in murine and human macrophages, and inhibition of TLR-4 by using its antagonist attenuates pro-inflammatory cytokine induction [17].

It was considered that pro-inflammatory cytokines greatly contribute to the pathogenesis of SARS-CoV-2 and its severity. Zheng et al., reported the sensing of SARS-CoV-2 E protein by Toll-like receptor-2 (TLR-2), which results in the hyper-expression of pro-inflammatory cytokines that may contribute to disease severity [18], suggesting that TLR-2 mediated inflammation plays a pathogenic role in SARS-CoV-2 infection.

Therefore, TLR signaling might be involved in the induction of pro-inflammatory mediators in COVID-19.

Recent evidences from current literature showed a central role of macrophages in SARS-CoV-2 infection [19,20]. Among inflammatory cells, macrophages play a crucial role in COVID-19 phlogistic mechanisms, in the induction and non-resolution, which subsequently lead to the persistence and amplification of the phlogistic state, culminating in death. Although monocyte blood cell count results in COVID-19 patients is often within the reference range, a high proportion of activated cells was been shown [21]. Patients with severe and critical COVID-19 disease, admitted to intensive care units (ICU), have elevated blood levels of IL-1β, IL-2, IL-7, IL-9, IL-10, IL-17, Granulocyte Colony-stimulating factor (G-CSF), Granulocyte-Macrophage Colony-Stimulating Factor (GM-CSF), Interferon-γ (IFN-γ), Tumor Necrosis Factor-α (TNF-α), C-X-C motif chemokine ligand 8 (CXCL8), C-X-C motif chemokine ligand 10 (CXCL10). Furthermore, fatal cases of SARS-CoV-1, MERS-CoV and SARS-CoV-2 lung autopsies have revealed extensive cellular infiltration of phlogistic cells with macrophages predominance [19].

Macrophages can actually sub-divided into three functional classes: M1, M2, and M2-like [22,23], surpassing the dichotomous classification in two classes of functionally distinct macrophages [24]. M1 macrophages are activated by intracellular pathogens, bacterial wall, lipoptoteins and many cytokines such as IFN-γ and TNF-α, while activation of M2 is induced by fungi, parasites, immune-complex, interleukins, and many other signals. M2 macrophages have phagocytosis capacity, but they can also mitigate the inflammatory response and promote the injury repair [25]. M2-like macrophage phenotype is under the control of IL-10, which in turn activates Signal Transducer and Activator of Transcription 3 (STAT3)-mediated gene expression of IL-10, Transforming Growth Factor beta 1 (TGF beta 1), Mannose Receptor C-type 1 (MRC 1) [26].

Regarding the anatomical sites, lung macrophages were been divided into three broad categories, since they are located in interstitial, alveolar, and in the vascular compartments respectively [27].

In the context of the alveolar macrophage population, the maintenance of the activation of pathway regulated by PPARγ (peroxisome proliferator-activated receptor gamma) has a fundamental role in the containment of lung inflammatory process, and in the resolution of acute phase, while on the contrary an important PPARγ repression was been demonstrated during activation of lung monocyte-macrophages in severe COVID-19 [28].

Interstitial macrophages also consist of heterogeneous populations, as reported in recent scientific articles [29,30].

An important role in the activation of M2 macrophage is played by Galectin-3 (GAL-3). Indeed, GAL-3 was been formerly called MAC-2 antigen because of its role in activating M2 macrophages [31].

In this study, we have analysed the immunohistochemical expression of TLR-4, GAL -3, CD68 and CD163 in the specimens derived from autopsy procedures of patients died from critical SARS-CoV-2 infection, in order to potentially associate innate immune hyperinflammation with monocyte-macrophage up-regulation and macrophage’s sub-population changes.

In details, aim of this work was to evaluate, in the complex context of macrophage activation, the pro-inflammatory role of TLR-4, and association among alveolar GAL-3 expression, innate immune phlogistic pathway activation, and lung macrophage repletion by CD163^(+)^ myeloid derived monocyte-macrophages.

Further aim is to evaluate possible association between a macrophage shift between loss of alveolar GAL -3 macrophages and persistence of Spike-1 viral sequence as a crucial event in lethal SARS-CoV-2 lung disease.

In summary, our study will focus on the innate-immune pathway, the inflammatory profile of macrophages, and composition of alveolar and interstitial infiltrate in lethal SARS-CoV-2 lung disease.

## 2. Results

### 2.1. Main Histopathological Findings

SARS-CoV-2 infection determined alveolar congestion, with pneumocytic damage, depicting a particular Diffuse Alveolar Damage (DAD) (Figure 1), representing the histological hallmark for the acute phase of Acute Respiratory Disease Syndrome (ARDS), together with acute phlogistic cell inflammation, interstitial vessel congestion, hemorrhage, and hyaline membrane deposition. Massive alveolar and interstitial phlogistic cell infiltration, with hyaline membrane, and thin fibril deposition was been detected, depicting a network of early-intermediate fibrosis. Furthermore, polymorphonuclear cells infiltrate the lungs, especially in cases associated with bacterial superinfection. Most deceased patients showed lungs already affected by chronic inflammation with lymphocytic infiltration and anthracotic deposits. SARS-CoV-2 infection determined in these lungs alveolar congestion with pneumocyte type I and type II damage. Proteinaceous fibrillar edema induced oxygen exchange impairment, leading to Acute Lung Injury, (ALI), and, in turn, ARDS.

### 2.2. In Situ Hybridization (ISH)

All patients who died from a lethal form of SARS-CoV-2 infection in our cohort of COVID-19 lethal disease had ARDS due to lung tissue viral persistence demonstrated by PCR as previously reported [32]. Therefore in this study we presented the percentage of Spike-1^(+)^ cases as evaluated by ISH compared to mean values ± Standard Error Means (SEM) of selected studied markers (TLR-4, CD68, CD163, GAL-3) (Table 1 and Figure 2).

### 2.3. TLR-4 Expression Was Up-Regulated in a Subgroup of Patients with Lethal COVID-19 Lung Disease

TLR-4 was strongly up-regulated in damaged pneumocytes in a subgroup of deceased COVID-19 patients (Figure 3A,B). Furthermore, TLR-4 was up-regulated in lung macrophages in a subgroup of deceased COVID-19 patients (Table 1 and Figure 3C,D). The average of TLR-4 stained macrophage per square area is up-regulated as compared to control cases, however, statistically significant values are not reached, in this cohort, due to heterogeneous variability of TLR-4 mediated hyper-inflammation in different deceased subjects. In detail, our digital pathology based immunohistochemistry showed a mean count of TLR-4 positive macrophages of 183.04 ± SEM 54.9 in COVID-19 patients, versus mean count of 42.67 ± SEM 10.2 in control cases. Nevertheless, our study showed a trend to TLR-4 up-regulation in lungs of a subgroup patients with lethal COVID-19, when compared to the control group of SARS-CoV-2 negative lungs. In particular, we showed a sub-group with strong TLR-4 over-expression (Figure 4).

### 2.4. GAL-3 Macrophages Depletion Is Associated with Persistence of Viral Spike-1 Sequence in COVID-19 Lethal Lungs

GAL-3 positive macrophages were up-regulated in Spike-1^(−)^ and significantly down-regulated in Spike-1^(+)^ SARS-CoV-2 infected lethal lungs respectively. Means and SEM were obtained by immunohistochemical expression of GAL-3 (clone 9C4) (Table 1 and Figure 5), as evaluated by digital pathology analysis; Spike-1^(+)^ status was detected by PCR-based methods and ISH (Figure 2). Immunohistochemistry showed a mean count in SARS-CoV-2^(−)^ controls of 29.67 ± 6.49 SEM, versus mean count of 47.00 ± 15.87 SEM, and 19.58 ± 9.81 SEM in Spike-1^(−)^ and Spike-1^(+)^ COVID-19 deceased patients respectively. Comparisons were statistically significant as evaluated by Mann-U-Whitney test (Table 1 and Figure 6).

### 2.5. GAL-3/CD68, and CD163/CD68 Ratios, and Their Role in Altering Balance between M2 and M1 Macrophages in the Lungs

Our data showed an increased pan-macrophage marker CD68 in COVID-19^(+)^ patients associated with imbalance of M2-like sub-populations. In detail, GAL-3/CD68 ratio, was down-regulated in SARS-CoV-2 infected lethal COVID-19 lung disease (Mean 14.92 ± 5 SEM), compared to controls (Mean 84.85 ± 24.9 SEM), as evaluated by Mann-U-Whitney test (*p* < 0.001) *(*Figure 7A). Furthermore, CD163/CD68 ratio was downregulated in SARS-CoV-2 infected lethal COVID-19 lung disease (Mean 19.9 ± 3.7 SEM), compared to controls (Mean 69.6 ± 14.47 SEM), as evaluated by Mann-U-Whitney test (*p* < 0.001) (Figure 7B).

### 2.6. TLR-4 Innate Immunity Marker Is Associated with GAL-3 Expression in SARS-CoV-2 Related Lethal Lung Disease

Expression of TLR-4 and GAL-3 positive macrophages, assessed by digital-pathology-immunohistochemical, showed statistically significant association by Spearman’s statistical correlation test (*p* < 0.001) (Figure 8).

### 2.7. CD68 Positive Macrophages Were Significantly Increased in COVID-19 Lethal Lungs

Our study showed a significant overexpression of pan-macrophage marker CD68 in lungs of deceased COVID-19 cases as compared to SARS-CoV-2 negative control lungs (Table 1). Up-regulation was observed irrespective of Spike-1 persistence. Means and SEM were obtained by immunohistochemical expression of CD68 (clone KP1) as evaluated by digital pathology analysis; Spike-1-^(+)^ status was detected by in ISH. In detail, digital pathology based immunohistochemistry demonstrated a mean count of CD68(+) macrophages in SARS-CoV-2 negative controls of 59.43 ± 20.42 SEM, versus mean count of 316.15 ± 34.63 SEM, and 314.92 ± 36.24 SEM in Spike-1 negative and Spike-1 positive COVID-19 deceased patients respectively (Table 1), comparison were significant (Figure 9).

### 2.8. Variation of the Macrophage Populations Found in the Lung of COVID-19 Deceased Patients Was Conditioned by the Infiltration of Myeloid Derived Monocytes-Macrophages as Evaluated by CD163

Mean and standard deviation was been obtained by immunohistochemical expression of CD-163 stained cells, as evaluated by digital pathology and Mann-Whitney U-test statistical analysis. At the time of the fatal event, a subgroup showing markedly high values was part of Spike-1 persistent lungs. Comparisons were not statistically significant, due to heterogeneous variability of CD163^(+)^ myeloid derived macrophages in different deceased subjects, but showed an interesting trend, as compared to the limited number of samples examined. In details, our study showed CD163^(+)^ mean count in SARS-CoV-2 negative controls of 34.33 ± 7.82 SEM, versus mean count of 39.15 ± 4.34 SEM, and 111.00 ± 36.65 SEM in Spike-1^(−)^ and Spike-1^(+)^ COVID-19 deceased patients respectively (Table 1 and Figure 10). Furthermore, CD163 and GAL-3 showed an inverse correlation, although not significant, when associated with Spearman’s test (Figure 11). This interesting trend requires confirmatory analysis on more representative samples.

## 3. Discussion

The involvement of TLRs in inflammation and bacterial infection have been recognized for a long time. There is an increasing number of reports demonstrating the involvement of TLR activation in a variety of viral infections, associated with protective immunity, but also immune hyper-activation and even viral replication.

Recent data show the involvement of TLR activation in various acute respiratory viral infections, including SARS-CoV-2 and indicate an essential role in COVID-19 pathology. A number of studies and meta-analyses conducted in the last 3 years have shown pulmonary involvement in the most of COVID-19 lethal cases [33,34]. However, the role played by monocyte-macrophage subpopulations requires further investigation, taking into account that macrophages were been recognized as a heterogeneous and dynamic population of cells that have the capacity to perform a wide range of critical functions [35].

During severe and critical SARS-CoV-2 infections, as well as in many severe infections leading to ARDS, macrophage destruction and depletion occur during alveolar infection with subsequent repopulation attempts by monocyte-marrow-derived macrophage populations. In response to viral infections, also without co-infections and even more in those with bacterial co-infections, the marrow production of cellular elements of the monocytic-macrophage series may shift towards a pro-inflammatory or immunoregulatory phenotype according to phase progression and interaction with host immune system. On this point, recent evidences suggest that the alterations of the immune responses under the control of monocytes-macrophages are certainly one of the elements affecting the severity of SARS-CoV-2 infections [36,37,38,39].

Patients with lethal COVID-19 associated with macrophage counts of >130/High Power Field (HPF) showed shorter survival time in an autopsy cohort studied by Cao W et al. [40].

Our study shows that in lethal cases of COVID-19 associated lung disease, the inflammatory cells that predominate in the histological picture are macrophages, together with lymphocytes and polymorphonuclear neutrophils. Another important result is the finding of a drastic depletion of GAL-3^(+)^ macrophages, associated with down-regulation of GAL-3^(+)^/CD-68^(+)^ ratio, and persistence of Spike-1 sequence.

GAL-3 has several effects in the immune response, especially in lung injuries. GAL-3 is highly expressed in endothelial and alveolar macrophages, playing an important role in mechanisms of lung repair and fibrosis [41]. GAL-3 has been associated with M2-like macrophage activation [42]. Indeed, GAL-3 previous name was MAC-2 antigen because of its role in the activating of M2 macrophages [31].

However, a recent study showed an increased blood concentration of GAL-3 in patients affected by COVID-19, associated with mortality and severity of patient affected by COVID-19 [43]. These apparently conflicting results may be possibly explained hypothesizing a reduction of tissue GAL-3 mediated by apoptosis, necroptosis and necrosis of tissue resident lung cells, together with up-regulation of serum levels of cleaved GAL-3 protein.

A series of evidence from current literature showed that GAL-3 antagonizes generation of a pro-inflammatory macrophage phenotype. In this context Di Gregoli K, et al., reported that the accumulation of GAL-3^(−)^ macrophage plays a central role to the heightened invasive capacity observed in the GAL-3^(−)^/CD-68^(+)^ macrophage sub-population in atherosclerosis. Additionally, they detected a clear shift toward a pro-inflammatory macrophage phenotype in response to GAL-3 silencing as mRNA expression of the pro-inflammatory molecules TNF-alpha, PTGS2 (prostaglandin-endoperoxide synthase 2 cyclooxygenase-2), and IL-6 was increased in response to GAL-3 depletion, implicating GAL-3 as a possible negative regulator of inflammation and proinflammatory macrophage polarization [44].

Our study demonstrates that in the lungs of subjects who died of critical forms of SARS-CoV-2 infection there is a drastic reduction of the GAL-3^(+)^ immune-regulatory macrophages associated with Spike-1 persistence, which instead should be responsible for promoting healing of acute inflammation, favoring the phases of chronic inflammation, and tissue repair-regeneration. These macrophages in subjects with co-morbidities consistently associated with the increased risk of mortality of the disease COVID-19 (neoplasms, COBP) are PD-L1+ and despite their role in the pathophysiology of these pathologies they still have a finalistic significance in reducing hyper-inflammation potentially lethal in the lungs. Therefore we demonstrate that the inability to generate an immunoregulatory phenotype in the lungs (PD-L1+, GAL-3+) is one of the main causes of lethality in the analyzed subjects.

The key innate immune sensing receptors are germ line-encoded PRRs, which mediate the initial sensing of infection by recognition of PAMPs upon microbial invasion of the host [4].

PRRs on the cell surface and endosomal membranes and in the cytosol may respond to SARS-CoV-2 PAMPs to activate innate immune signaling pathways. TLR-11, TLR-2, TLR-4 and TLR-6 can signal through MyD88 to activate Nucelar Factor kappa-light-chain-enhancer of activated b cells (NF-κB) and Mitogen-activated Protein Kinase (MAPK) signaling pathways to induce transcription of genes encoding pro-inflammatory cytokines and other sensors [45]. TLR-3 and TLR-4 can signal through TIR domain containing adapter-inducing interferon (TRIF) to activate Interferon 3 (IRF3), and induce expression of type I and type III IFNs. Strong experimental evidence supports SARS-CoV-2-mediated activation of TLR-22, while activation of TLR-1, TLR-3, TLR-4 and TLR-6 has been suggested bioinformatically and through associative studies [46].

Interestingly, Quero, L et al., showed that TLR-22 stimulation by Pam3 impairs anti-inflammatory activity of M2-like macrophages, generating a chimeric M1/M2 phenotype [47]. In summary, these results indicate that TLR-2, and to a higher degree TLR-4, ligands are able to change the anti-inflammatory M2 gene markers toward an M1-specific expression phenotype. CD14 and CD163 marker expression on M2 macrophages did not change upon TLR-2 and TLR-4 engagement. By contrast, M2 gene markers *HMOX1*, *FOLR2*, and *SLC40A1* were decreased phenotype [47]. NF-κB was, similarly, activated in M1 and M2 by TLR2 and TLR-4 ligands. Thus, following TLR-2 stimulation by its ligand Pam3, the M2 population secreted the pro-inflammatory cytokines IL-6 and IL-8 at levels comparable to Pam3-stimulated M1-polarized macrophages. Therefore, Quero, L et al. concluded that despite this shift toward a proinflammatory M1 function, M2 macrophages continued to express the typical M2 cell surface markers CD14 and CD163 phenotype [47].

Furthermore, we hypothesize that according to Quero L, et al. [47], in deceased patients from this cohort, despite a myeloid response directed to M2 macrophage recovery function, with expression the typical M2 phenotype and cell surface markers CD163, there is a functional pathological shift toward a persistent pro-inflammatory M1 state.

On this topic, recent results support that timely acquisition of a myeloid cell immune-regulatory phenotype might contribute to recovery in severe systemic SARS-CoV-2 infection and suggest that therapeutic agents favoring an innate immune system regulatory shift may represent the best strategy to implement at this stage [48]. Furthermore, on this topic Grassi G. et al. reported a paradoxical effect of Myeloid-Derived Suppressor Cells in COVID-19 [49].

On the basis of experimental data, in our cohort of lethal cases, we demonstrate that the inflammatory microenvironment induced by SARS-CoV-2 infection through innate immunity molecules determines an hyper-inflammatory state capable of influencing the presence of functionalized macrophages in the M1- direction with overexpression of TLR-4, and that, at the same time, macrophages with the M2-like CD-163^(+)^ phenotype persist in the alveolar and interstitial compartments. Furthermore, M2-like activity could be further impaired by TLRs overexpression in local alveolar and interstitial compartments, due to cytokine storm, leading to unresolved inflammation and death.

Expression of TLR-4^(+)^ and GAL-3^(+)^ stained macrophages, assessed by digital-pathology-immunohistochemistry, showed statistically significant association by Spearman’s statistical correlation test (*p* < 0.001).

This study demonstrated, in the complex context of macrophage activation, association between GAL-3 expression and innate immune phlogistic pathway activation in patients deceased from SARS-CoV-2 infection.

Further results deriving from our research are the demonstration of a macrophage sub-population shift, and association between loss of alveolar GAL-3 macrophages and persistence of viral sequences Spike-1 as a crucial event in lethal COVID-19 lung disease.

In addition, our study focuses on the variation of pulmonary inflammatory infiltration by bone myeloid-derived monocyte-macrophages, as demonstrated by immunolabeling with CD163.

This study also represents the histological basis for hypothesizing that early, prolonged and dose appropriate therapy with drugs such as high power steroids capable of modulating the innate inflammatory response can potentially reduce the mortality associated with SARS-CoV-2 infection. On this topic, in a recent systematic review and meta-analysis, Chaudhuri D. et al. showed that the use of corticosteroids reduces mortality in patients with ARDS. This effect was consistent between patients with COVID-19, and non-COVID-19 ARDS [50]. Furthermore, translational research in ARDS patients randomized to methylprednisolone has demonstrated the ability of corticosteroid therapy to rescue the cellular concentrations and functions of activated Glucocorticoid Receptors GRα (GC-GRα) leading to downregulation of systemic and pulmonary NF-κB-activated markers of inflammation, coagulation, and fibroproliferation [51,52,53]

## 4. Materials and Methods

### 4.1. Ethics Approval

The study was conducted according to the guide-lines of the Declaration of Helsinki and approved by the Attorney’s Office of Naples, protocol number p.p. 509568/020/44, dated 15 May 2020.

### 4.2. Clinicopathological Data of Selected Patients

A series of 25 patients, the most (24/25, 96%) with co-morbidities (NIDDM, Type II-Non-Insulin-Dependent Diabetes Mellitus; OCBP, Obstructive Chronic Broncho-Pneumopathy; CN, Chronic Nephropathy; O, Obesity; H, Hypertension; CH, Cardiac Hypertrophy), who died from a lethal form of SARS-CoV-2^(+)^ infection (COVID-19^(+)^), of which 13 males and 12 females, aged between 45 and 80 years were studied. The study was conducted exclusively on formalin fixed paraffin embedded (FFPE) lung tissues. Lung tissues of the cohort of COVID-19^(+)^ patients were compared to SARS-CoV-2^(−)^ lung tissues selected from a pool of controls constituting the COVID-19-negative cohort, deriving from lung surgery specimens for primary or metastatic lung tumors, on tumor-free margins and at least 3 cm distant from the latter, or from surgery specimens for non-tumor-related lung surgery. All the surgical procedures were conducted with curative intent, according to standard approved therapeutic protocols. Clinicopathological data, and details regarding co-morbidity/morbidity of both COVID-19^(+)^ and COVID-19^(−)^ cohorts has been collected (Table 2). The control study collection was conducted over a three-year period (2020–2023), and due to the limited amount of lung tissue available, comparisons between COVID-19^(+)^ and COVID-19^(−)^ cases were gradually conducted not on all controls, but on a minimum number of at least 6 of the overall total specimens available (total 11 cases).

Autopsy cases were been collected between June 2020 and July 2021 at time of wild-type SARS-CoV-2 with early onset of Epsilon and Alpha variants [54].

Patients deceased in a mean interval between diagnosis and death of 62.52 days (minimum of 20 days, maximum 120 days).

### 4.3. Autopsy Protocol

Deceased patients underwent autopsy, through help of a ventilation system with six complete air changes/h (ACH) in a pressure-negative environment, with air exhausted through HEPA filters [Biosafety Level 3 (BSL3)], according to the University of Padova autopsy protocol [55,56].

Specimens of lung tissue from 25 patients deceased for SARS-CoV2 were been used as representative cases of COVID-19 lethal ARDS.

### 4.4. Methods

Immunohistochemical analysis has been performed on formalin-fixed paraffin embedded specimens derived from autopsy procedures (COVID-19^(+)^ cohort), and from routine lung surgical operations (COVID-19^(−)^ cohort). Single immunostaining has been performed using specific monoclonal antibodies against TLR-4 (clone 76B357.1), GAL-3 (clone 9C4), CD68 (clone KP1), and CD163 (Clone MRQ-26), revealed by standard linked-streptavidin-biotin immunoperoxidase technique (LSAB), developed by 5′diaminobenzidine and/or alkaline phosphatase methods (Table 3).

Evaluation was been performed by two blinded observers analyzing standard High Power Field (HPF) area, with 20× magnification, measuring 45,508.61 µm^2^. At least eight HPFs were been evaluated for a mean total area of 1,160,000 µm^2^. Antibodies used and experimental conditions have been reported in Table 3.

The percentage of positive immunostained cells was been evaluated by digital pathology analyses using cellSens Dimension and Image-Pro Premier Offline (MediaCybernetics, Version 9.1.4 Build 5638) software.

All the controls have pathological morbidity leading to surgical intervention for therapeutic purpose. In details, two out eleven cases had metastasis to the lung from distant carcinomas (one from breast cancer and one from large bowel cancer), six out 11 cases had primary lung adenocarcinomas, one out 11 had lung hamartoma, and two out 11 had complicated lung emphysema. This COVID-19^(−)^ cohort, carrying slight chronic lung pneumonia, have been used to match samples of SARS-CoV-2^(+)^ infected hyper-inflamed lethal cases of pneumonia, leading to not resolved COVID-19 related ARDS and death, representing the COVID-19^(+)^ cohort.

### 4.5. In Situ Hybridization (ISH) Method for SARS-CoV-2 Detection

Briefly, ISH detection of SARS-CoV-2 was been performed using V-nCoV2019-S probe to detect viral sequences Spike (S) of SARS-CoV-2 in lung tissue using RNAscope 2.5 high-definition detection kit (Advanced Cell Diagnostics). SARS-CoV-2 was been detected in lung tissues of lethal COVID-19 disease by in ISH according to methods reported in previous work of our laboratories in which we compared ISH with PCR methods [32,56,57], and in current literature [58,59].

### 4.6. Statistical Analyses

Data have been analyzed by SOFA Statistics 1.4.6, and SPSS 28.01.0 Data Analysis and Statistical Software and Windows Operating Systems; Spearman’s test and ANOVA statistical analysis were been used to correlate to clinicopathological parameters. Mann-U-Whitney test was been used to compare non-parametrical variables (Table 1).

## 5. Conclusions

In summary, our study focused on the innate-immune pathway, the inflammatory profile of macrophages and composition of alveolar and interstitial infiltrate in lethal SARS-CoV-2 lung disease.

Our study shows that in lethal cases of COVID-19 associated lung disease, the inflammatory cells that predominate in the histological picture are TLR-4^(+)^ damaged pneumocytes, with phlogistic infiltration of macrophages, together with lymphocytes and polymorphonuclear neutrophils. Another important result is the finding of a drastic depletion of GAL-3^(+)^ macrophages, associated with down-regulation of GAL-3^(+)^/CD-68^(+)^ ratio, and persistence of Spike-1 sequence.

The association between loss of alveolar GAL-3 macrophages and persistence of viral sequences is a crucial event in lethal COVID-19 lung disease in the cohort of analyzed cases.

In this study we showed that inflammatory factors of the pulmonary microenvironment can condition an adverse and lethal response despite the triggering of a bone marrow response potentially favorable to healing. The inflammatory microenvironment induced by SARS-CoV-2 infection through innate immunity molecules determines an hyper-inflammatory state capable of influencing the presence of functionalized macrophages in the M1 direction with overexpression of TLR-4, and that, at the same time, macrophages with the M2-like CD-163^(+)^ phenotype persist in the alveolar and interstitial compartments. Furthermore, M2-like activity could be further impaired by TLRs overexpression in local alveolar and interstitial compartments, due to cytokine storm, necroptosis, and microthrombosis, in turn leading to unresolved inflammation and death.

This study also represents the histological basis for hypothesizing that early, prolonged and dose appropriate therapy with drugs such as high power steroids capable of modulating the innate inflammatory response can potentially reduce the mortality associated with SARS-CoV-2 infection.

## Figures and Tables

**Figure 1 ijms-24-13259-f001:**
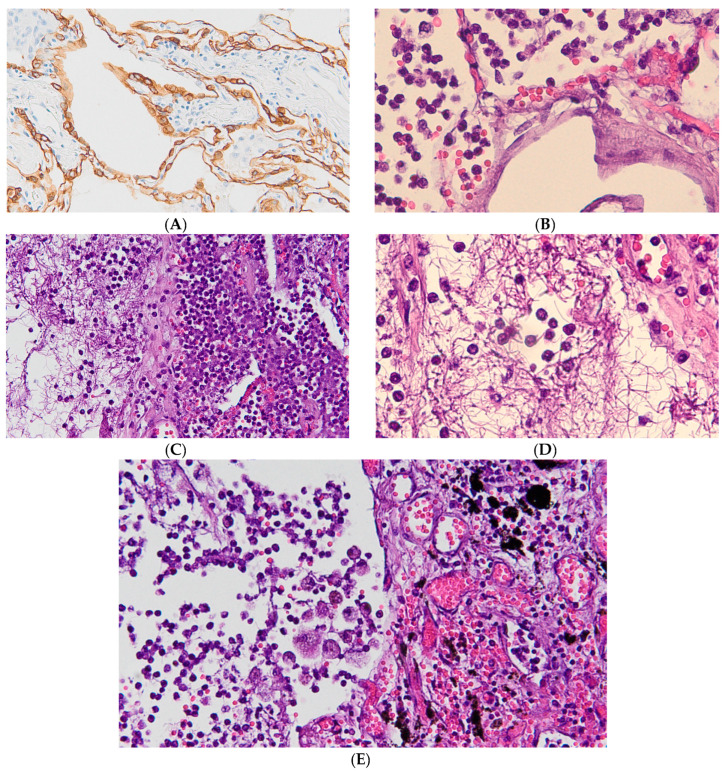
Histopathological picture of lungs in SARS-CoV-2^(+)^ lethal infection (**B**–**E**) compared to SARS-CoV-2^(−)^ control (**A**). (**A**) Type I and type II pneumocytes are major target of SARS-CoV-2 infection. This picture overview shows human intact pneumocytes stained with pan-cytokeratin marker (clone CK-AE1/AE3/PCK26) by standard immunohistochemistry, and slight interstitial lymphocytic inflammation (unstained cells) in a representative SARS-CoV-2^(−)^ control patient (H&E original magnification, ×20 HPF). (**B**) SARS-CoV-2^(+)^ infection determined alveolar congestion with pneumocyte damage, acute phlogistic cells inflammation, haemorrage, as expression of early stage of Diffuse Alveolar Damage (H&Ε staining, original magnification, ×40 HPF). (**C**) Massive alveolar and interstitial phlogistic cell infiltration with membrane hyaline and thin fibril deposition, depicting a network of early fibrosis (H&Ε staining, Original magnification, ×20). (**D**) Massive alveolar and interstitial phlogistic cell infiltration with hyaline membrane and thin fibril deposition. Note interstitial vessel congestion and slight haemorrage. In this case, polymorphonucleates infiltrate lungs, this was been especially observed in cases with Gram-negative bacterial superinfection (H&E, original magnification, ×40 HPF). (**E**) SARS-CoV-2 lethal damage in previous damaged lungs. The lung was been already affected by chronic inflammation with anthracotic deposits (H&E, original magnification, ×40 HPF).

**Figure 2 ijms-24-13259-f002:**
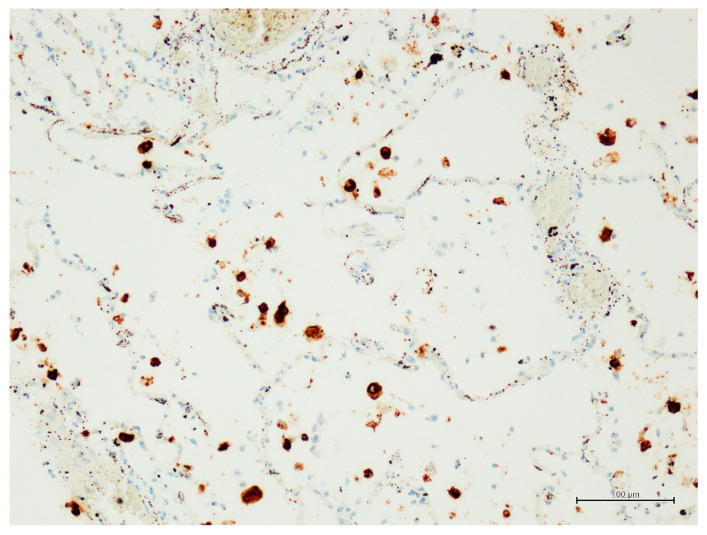
In Situ Hybridization (ISH) for Spike-1 of SARS-CoV-2 in infected human lungs. Representative case of ISH for Spike-1 viral strand in lung alveolar macrophages of COVID-19 deceased case [ISH detection of SARS-CoV-2 has been performed using V-nCoV2019-S probe detect viral sequences of Spike (S) of SARS-CoV-2 in lung tissue using RNAscope 2.5 high-definition detection kit (Advanced Cell Diagnostics), Original magnification, ×20].

**Figure 3 ijms-24-13259-f003:**
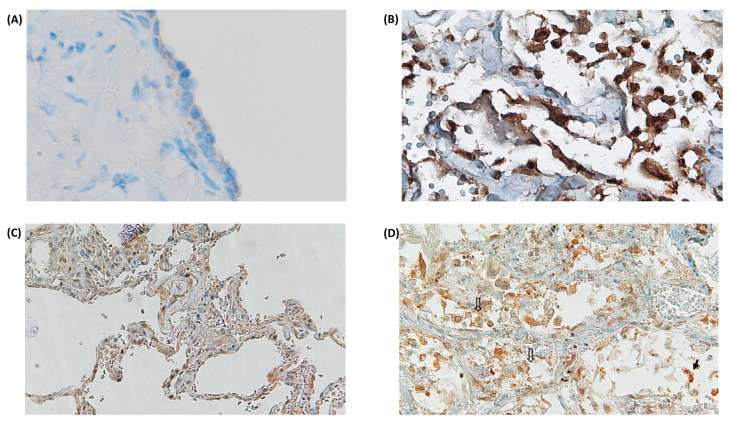
Over-expression of TLR-4 in SARS-CoV-2 injured lungs. Note up-regulation in SARS-CoV-2 damaged lung pneumocytes (**B**), as compared to human normal pneumocytes (**A**), and in pneumocytes and macrophages (**C**) as compared to SARS-CoV-2 negative lung with emphysema and slight interstitial chronic infiltration (**D**) [(↓) Solid black arrows indicate damaged pneumocytes, and (⇓) empty arrows indicate macrophages; standard LSAB-technique, nuclear counterstaining with haematoxylin, original magnification ×20].

**Figure 4 ijms-24-13259-f004:**
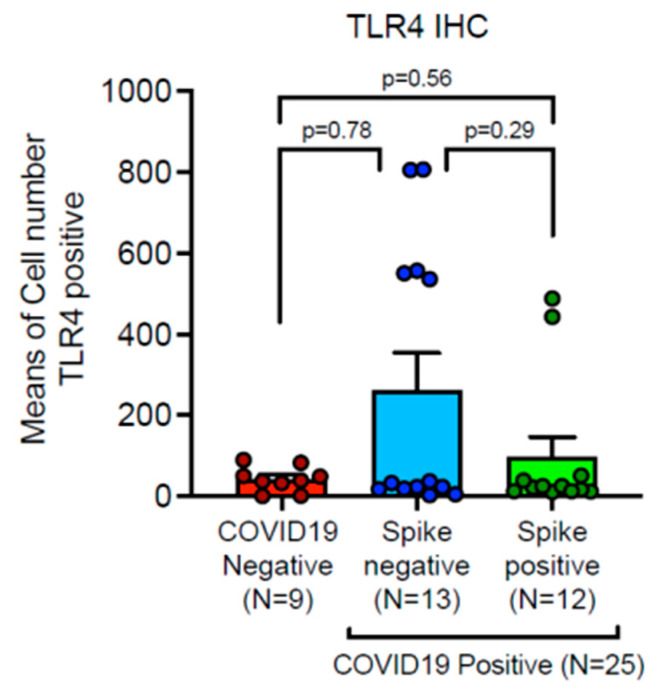
Up-regulation of TLR-4 in a subgroup of lethal COVID-19 lung disease—statistical comparison. Expression of TLR-4 showed a trend to up-regulation in lungs of a subgroup patients with lethal COVID-19, when compared to the control group of SARS-CoV-2 negative lungs. Note that a subgroup of patients demonstrates significant up-regulation of TLR-4 innate-immunity marker, irrespective of Spike-1 persistence. Mean ± SEM were obtained by immunohistochemical expression of TLR-4 (CD 284), as evaluated by digital pathology analysis (Mann-Whitney U-test).

**Figure 5 ijms-24-13259-f005:**
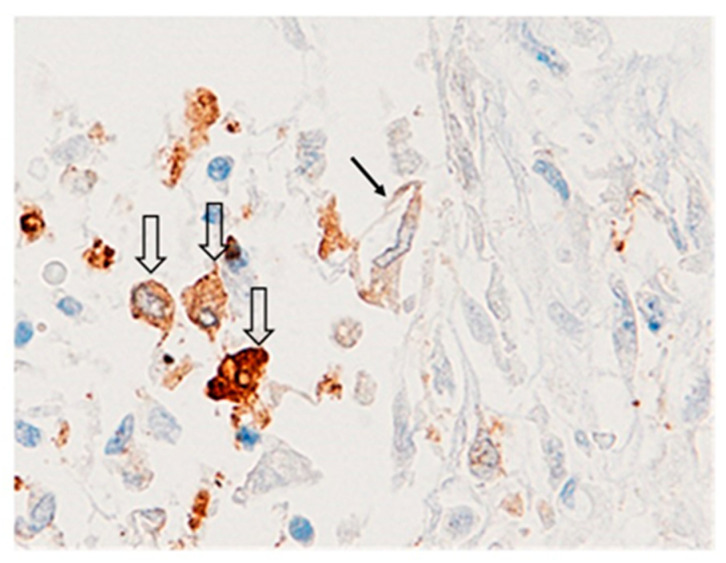
GAL-3 immunostaining with standard LSAB-HRP in representative case of SARS-CoV-2 infected lung in a patient deceased in the early stage of ARDS. The picture was captured in hot spot for GAL-3 in section serially stained with Spike-1 sequence by ISH. Note the persistence of GAL-3 expression in macrophages and damaged pneumocytic cells [(↓) Solid black arrows indicate damaged pneumocytes, and (⇓) empty arrows indicate macrophages; LSAB performed with mononuclear Ab against GAL-3, nuclear counter stainig with Gill’s type-II Haematoxylin; original magnification ×20].

**Figure 6 ijms-24-13259-f006:**
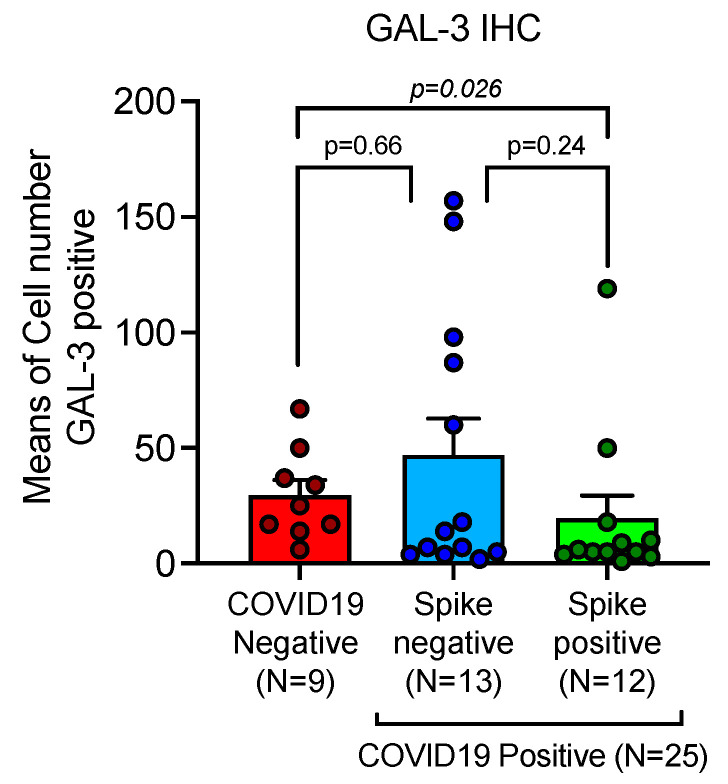
GAL-3 positive macrophage are up-regulated in Spike-1^(−)^ and significantly down-regulated in Spike-1-^(+)^ SARS-CoV-2 infected lethal lungs. GAL-3 positive macrophage infiltration in SARS-CoV-2 shows a trend of increased levels in Spike^(−)^ COVID-19 cases, while a significant down-regulation associated with the presence of viral Spike-1 sequence was demonstrated; means and SEM were obtained by immunohistochemical expression of GAL-3 (clone 9C4), as evaluated by digital pathology analysis; Spike^(+)^ status was detected by PCR-based methods and ISH; Mann-Whitney U-test (*p* < 0.05).

**Figure 7 ijms-24-13259-f007:**
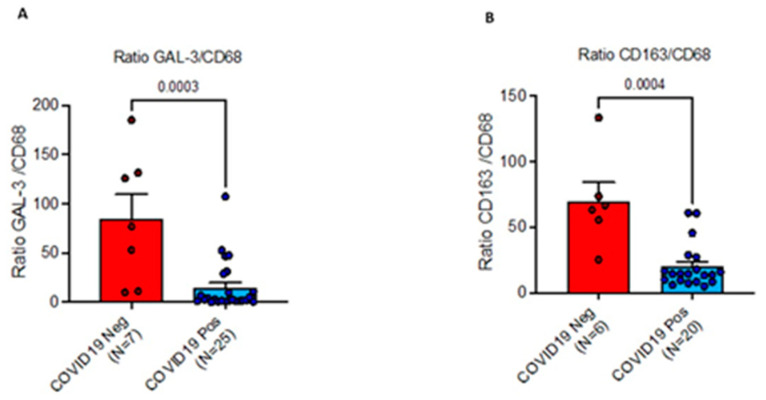
(**A**) GAL-3/CD68 ratio in COVID-19^(+)^, compared to COVID-19^(−)^, as evaluated by Mann-Whitney U-test (*p* < 0.001). These data show increase of pan-macrophage marker CD68 associated with imbalance of M2-like macrophage sub-population infiltrating alveolar and interstitial compartments of SARS-CoV-2^(+)^ lethal lung disease. (**B**) CD163/CD68 ratio in COVID-19^(+)^, compared to COVID-19^(−)^, as evaluated by Mann-Whitney U-test (*p* < 0.001). CD163/CD68 ratio was downregulated in SARS-CoV-2 infected lethal COVID-19 lung disease, compared to controls as evaluated by Mann-U-Whitney test (*p* < 0.001).

**Figure 8 ijms-24-13259-f008:**
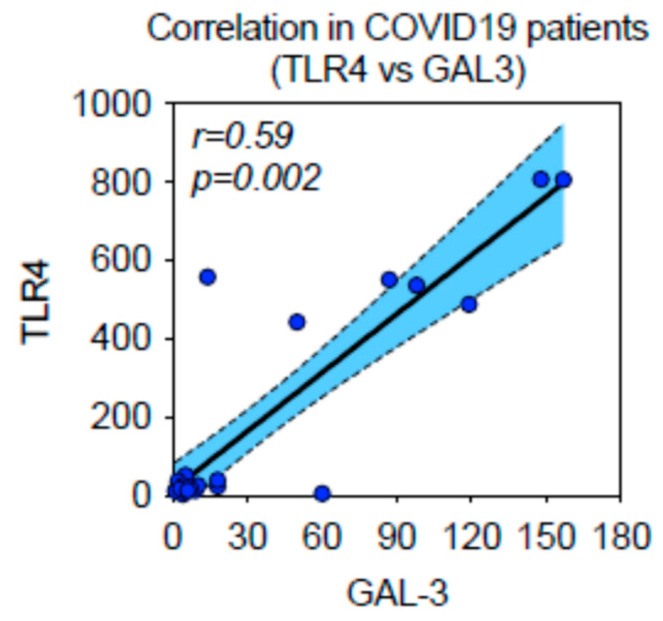
TLR-4 innate immunity marker is associated with GAL-3 expression in SARS-CoV-2 related lethal lung disease. Expression of TLR-4 and GAL-3 positive macrophages, assessed by digital-pathology-IHC, showed statistically significant association by Spearman’s statistical correlation test (*p* < 0.01).

**Figure 9 ijms-24-13259-f009:**
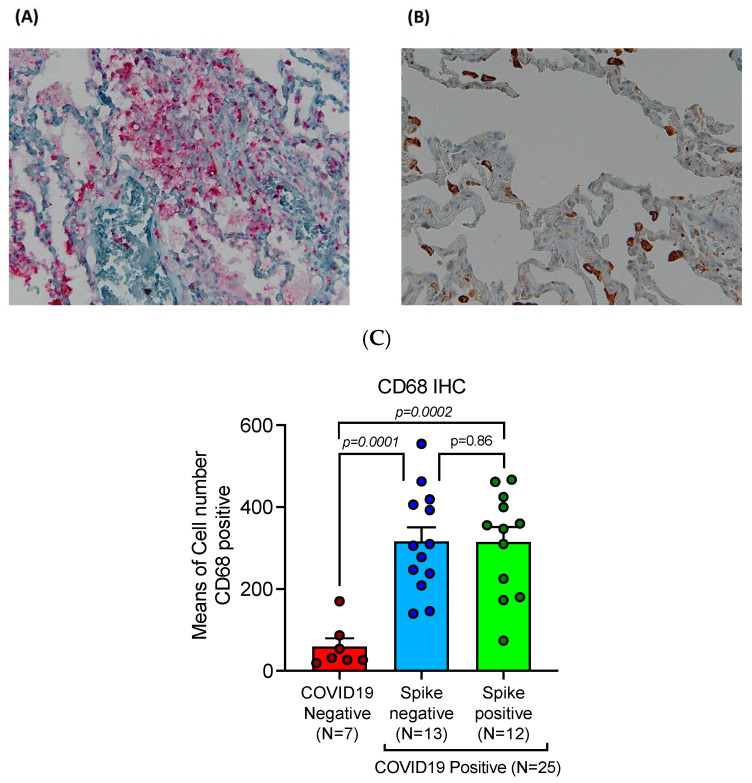
Endoalveolar macrophages exudation as evaluated by CD-68 immunostaining in COVID-19 related lungs (**A**) and SARS-CoV-2 unrelated lung (**B**) (original magnifications, ×20); statistical comparisons (**C**): the increase in the total number of macrophages was associated with SARS-CoV-2 infection with statistically significant values in both groups of Spike-S1^(+)^ and Spike-S1^(−)^ cases, when compared to control group. Means and SEM were obtained by immunohistochemical expression of CD68 (clone KP1), as evaluated by digital pathology analysis; Spike-1-positive status was detected by in ISH; Mann-U-Whitney test (*p* < 0.001).

**Figure 10 ijms-24-13259-f010:**
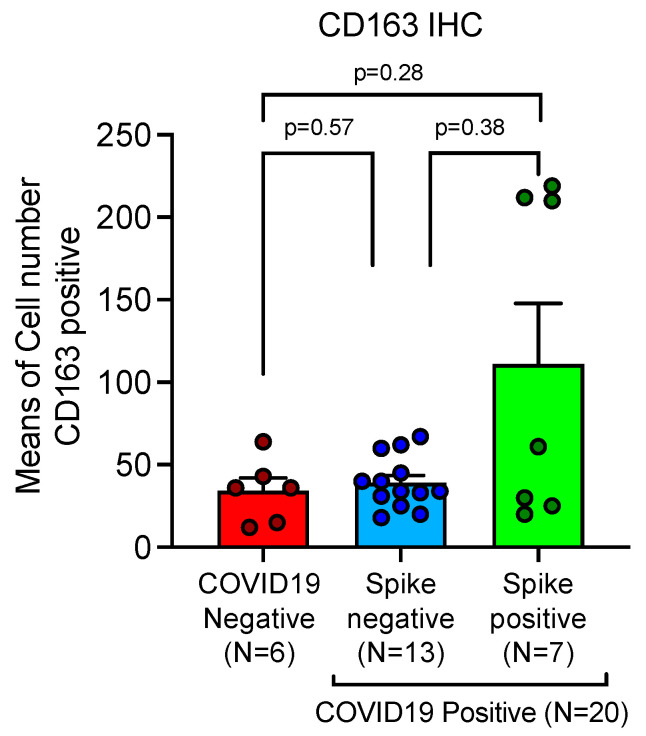
Association between up-regulation of CD-163 macrophage marker and Spike-1 detection in lethal COVID-19 lung disease. Expression of CD163 showed up-regulation in lungs of patients with lethal COVID-19, when compared to the control group of SARS-CoV-2 negative lungs. At the time of the fatal event, a subgroup showing markedly high values was part of Spike-1 persistent lungs. Mean and SEM were obtained by immunohistochemical expression of CD163, as evaluated by digital pathology and statistical unpaired *t*-test/Mann-Whitney U-test analysis. Comparisons were not statistically significant, but showed an interesting trend, in relation to the limited number of samples examined.

**Figure 11 ijms-24-13259-f011:**
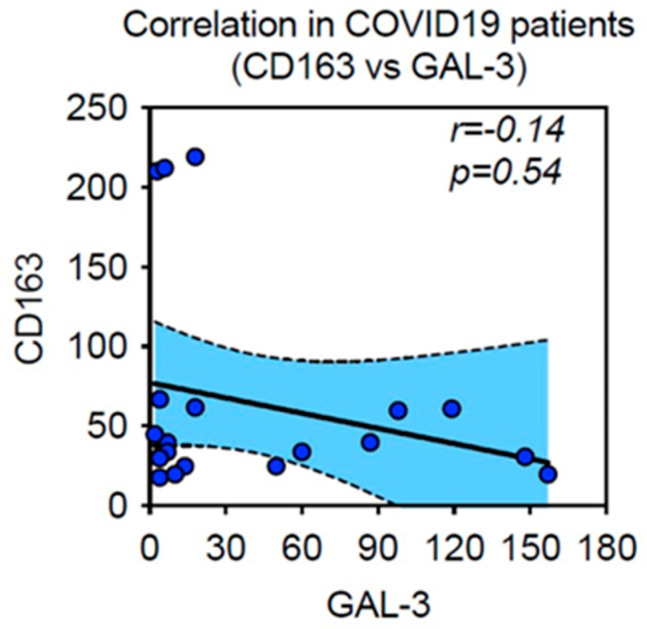
Association between CD163 and GAL-3 in lethal COVID-19 lung disease. CD163 and GAL-3 showed an inverse relationship (r = −0.14), even though not significant, when compared with Spearman’s correlation test.

**Table 1 ijms-24-13259-t001:** Summary of Mean ± SEM of TLR4, CD68, CD163, and GAL-3 immunoreactivity values according to COVID-19 status, and Spike-1 ISH detection.

COVID-19 Status	Immunohistochemistry				Spike-1 (ISH) Positive/Total, (%)
	TLR-4 Mean ± SEM	CD68 Mean ± SEM	CD163 Mean ± SEM	GAL-3 Mean ± SEM	
Positive	183.04 ± 54.90	314.92 ± 36.24	111.00 ± 36.65	19.58 ± 9.81	Spike-1 positive 12/25(48)
		316.15 ± 34.63	39.15 ± 4.34	47.00 ± 15.87	Spike-1 negative 13/24(52)
Negative	42.67 ± 10.20	59.43 ± 20.42	34.33 ± 7.82	26.67 ± 6.49	

**Table 2 ijms-24-13259-t002:** Summary of clinic-pathological parameters of patients groups according to COVID-19 status.

Variables	COVID-19 Status	
	Positive (Total n = 5)	Negative (Total n = 11)
Age, Mean ± SEM	65.76 ± 2.11	68.5 ± 5.40
Gender, Male/Female (%)	13/12 (1.08)	8/3 (2.66)
Pathological co-morbidities, Yes/Not (%)	24/1 (96)	11/11 (100)
Single co-morbidity/morbidity, (%)	12/25 (48)	11/11 (100)
Type of single co-morbidities/morbidity, (%)	Hypertension 18/25 (72)	Lung metastatic carcinoma 2/11 (18.18)
	Cardiac Hypertrophy 8/25 (32)	Primary lung neoplasm 6/11 (54.54)
	OCBP, 3/25 (12)	Lung emphysema 2/11 (18.18)
	Chronic nephropathy 2/25 (8)	Lung hamartoma 1/11 (9.09)
	Obesity 5/25 (20)	0/11 (0)
Multiple co-morbidities/morbidity, (%)	12/25 (48)	0/11(0)
Type of multiple co-morbidities/morbidity, (%)	Hypertension; CH, Cardiac Hypertrophy 6/25 (24)	
	Obesity, and NIDDM 2/25 (8)	
	Obesity, hypertension, and NIDDM 1/25 (4)	
	Hypertension; CH, Cardiac Hypertrophy, and Chronic nephropathy 7/25 (28)	
	Obesity, Hypertension, and Cardiac Hypertrophy 1/25 (4)	

Legend. NIDDM, Type II-Non-Insulin-Dependent Diabetes Mellitus; OCBP, Obstructive Chronic Broncho-Pneumopathy; CH, Cardiac Hypertrophy.

**Table 3 ijms-24-13259-t003:** Antibodies used, and experimental conditions.

Antibody	Clone	Method
TLR-4	76B357	LSAB-HRP/AP, Ventana Benchmark^®^ XT autostainer
CD68	KP1	LSAB-HRP/AP, Ventana Benchmark^®^ XT autostainer
CD163	MRQ-26	LSAB-HRP/AP, Ventana Benchmark^®^ XT autostainer
GAL-3	9C4	LSAB-HRP/AP, Ventana Benchmark^®^ XT autostainer

## Data Availability

Not applicable.

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
