# Peer review of "The Role of Toll-like Receptor-4 in Macrophage Imbalance in Lethal COVID-19 Lung Disease, and Its Correlation with Galectin-3"

_ijms, 2023, doi:10.3390/ijms241713259_

Round 1
Reviewer 1 Report
In this manuscript deceased COVID-19 patients are examined on histopathological lungs in order to quantify Gal-3 and TLR-4 expression in myeloid-derived cells by PCR and in situ hybridisation. Pan-macrophage marker CD68, myeloid derived monocyte-macrophages marker CD163, as well as four viral sequences served for capturing macrophage shift and viral infection in the samples.
While the pathological expertise is undoubted, there seem to be some inconsistencies in figures and text, which need clarifications and tidying up. Please have a look to the listed points below for specifictions. Furthermore, if no significant difference is shown, the interpretation should be set more carefully. Last but not least, language issues are found (some examples see below), please do a check on this.
Major issues:
- line 39: TLR-4 up-regulation is only seen in a subgroup of patients, which is even not the majority of patients.
- lines 154-165: Descriptive text is lacking supporting figures.
- Fig.2 C+D => C shows more TLR-4 staining than D, and C is the SARS-CoV-2 negative one? Then statement lines 174f "TLR-4 was up-regulated in lung macrophages of deceased COVID-19 patients (Figure 2 C- D)." does not fit.
- line 171: "TRL-4 expression was up-regulated in patients with lethal COVID-19 lung disease" => "TLR-4 expression was upregulated in a subgroup of patients with lethal COVID-19 lung disease"
- In the results section it would be good to join very short paragraphs/subsection and find new subheadings accordingly.
- Maybe it would help to see the correlation CD163 vs GAL-3?
- Fig.2: TLR-4 high and COVID-19 positive patients: characterisation/distribution of the subgroup to female/male, >50/<50 years age? Maybe it would help to perform the statistical analysis on these sample separately to controls in order to get a significant difference?
- Fig.9+10: If Fig.10 is just a more specific resolution of Fig.9, then Fig.9 is not needed. Furtheron, 20 COVID-19 patients were examined only in contrast to all other figures with 25. Is there a reason for this? Some figures include 9, 7, 6 controls, although 12 control samples are described in the method section. Why is that so?
- lines 465ff: Where are the results for all four viral RNA expressions?
Minor issues:
- Abstract: Since the abstract is separated by the subtitles and paragraphs, the numbers are superfluous.
- lines 44f: Direct molecular mechanism of TLR-4 on inflammation is not shown in this paper, so it should be phrased similar to: "Data show that TLR-4 expression correlates with a persistent activation of inflammation,"
- line 61: "including (TLRs)." => "including TLRs."
- Reduce paragraphs in the introduction. Many single sentences or two sentences are set in a single paragraph, which separation does not seem necessary.
- line 140: "In details. aim of this work was to evaluate," => "In detail, the aim of this work was to evaluate,"
- lines 151-153: Text can be dropped, since this is obvious and structured as usual.
- Results section: subheadings without full stop at the end, eg line 154 "Main histopathological findings." => "Main histopathological findings"
- line 174: "macropage" => "macrophage"
- line 176: "coorte" => "cohort"
- line 184: "Gal-3 macrophages depletion is significantly associated" => "Gal-3 macrophages depletion is associated"
- Fig.2: C + D double
- Fig.3, lines 252f: "Neg, Negative Covid-19 status; Pos, Positive Covid-19 status;" => can be omitted, since this is given in figure itself
- Fig.4, line 255: "Galectin-3 positive macrophage are up-regulated" => "Galectin-3 positive macrophages are up-regulated"
- Fig.4, lines 261ff: "Neg, Negative Covid status; S neg, COVID-19 status positive and Spike protein negative; S pos, COVID-19 status positive and Spike-1 protein positive;" => can be omitted, since this is given in figure
- Fig.5: It would help to insert arrows to indicate macrophages and damaged pneumocytic cells in different colours in order to help non-experienced readers.
- Consistent writing eg line 265 "Sars-CoV2" => "SARS-CoV-2"; Fig.3: "COVID19" => "COVID-19"
- line 268: "machrophages" => "macrophages"
- Fig.6: maybe easier to read, if figure x-axis label/description as in Fig.3 => Negative / Positive; COVID-19 status
- Results section: lines 155-170 => Figure connections are missing.
- line 444: "someone comorbidities"? Whole sentence is unclear.
- table 1: What do the rows mean "COVID-19 (N.1)", "Control Cases (N.0)" etc for the age distinction?
Some language issues are found, eg:
- line 174: "macropage" => "macrophage"
- line 176: "coorte" => "cohort"
- Consistent writing eg line 265 "Sars-CoV2" => "SARS-CoV-2"; Fig.3: "COVID19" => "COVID-19"
- line 268: "machrophages" => "macrophages"
Author Response
Major issues:
- line 39: TLR-4 up-regulation is only seen in a subgroup of patients, which is even not the majority of patients.
- lines 154-165: Descriptive text is lacking supporting figures.
- Fig.2 C+D => C shows more TLR-4 staining than D, and C is the SARS-CoV-2 negative one? Then statement lines 174f "TLR-4 was up-regulated in lung macrophages of deceased COVID-19 patients (Figure 2 C- D)." does not fit.
- line 171: "TRL-4 expression was up-regulated in patients with lethal COVID-19 lung disease" => "TLR-4 expression was upregulated in a subgroup of patients with lethal COVID-19 lung disease"
- In the results section it would be good to join very short paragraphs/subsection and find new subheadings accordingly.
- Maybe it would help to see the correlation CD163 vs GAL-3?
- Fig.2: TLR-4 high and COVID-19 positive patients: characterisation/distribution of the subgroup to female/male, >50/<50 years age? Maybe it would help to perform the statistical analysis on these sample separately to controls in order to get a significant difference?
- Fig.9+10: If Fig.10 is just a more specific resolution of Fig.9, then Fig.9 is not needed. Furtheron, 20 COVID-19 patients were examined only in contrast to all other figures with 25. Is there a reason for this? Some figures include 9, 7, 6 controls, although 12 control samples are described in the method section. Why is that so?
- lines 465ff: Where are the results for all four viral RNA expressions?
Minor issues:
- Abstract: Since the abstract is separated by the subtitles and paragraphs, the numbers are superfluous.
- lines 44f: Direct molecular mechanism of TLR-4 on inflammation is not shown in this paper, so it should be phrased similar to: "Data show that TLR-4 expression correlates with a persistent activation of inflammation,"
- line 61: "including (TLRs)." => "including TLRs."
- Reduce paragraphs in the introduction. Many single sentences or two sentences are set in a single paragraph, which separation does not seem necessary.
- line 140: "In details. aim of this work was to evaluate," => "In detail, the aim of this work was to evaluate,"
- lines 151-153: Text can be dropped, since this is obvious and structured as usual.
- Results section: subheadings without full stop at the end, eg line 154 "Main histopathological findings." => "Main histopathological findings"
- line 174: "macropage" => "macrophage"
- line 176: "coorte" => "cohort"
- line 184: "Gal-3 macrophages depletion is significantly associated" => "Gal-3 macrophages depletion is associated"
- Fig.2: C + D double
- Fig.3, lines 252f: "Neg, Negative Covid-19 status; Pos, Positive Covid-19 status;" => can be omitted, since this is given in figure itself
- Fig.4, line 255: "Galectin-3 positive macrophage are up-regulated" => "Galectin-3 positive macrophages are up-regulated"
- Fig.4, lines 261ff: "Neg, Negative Covid status; S neg, COVID-19 status positive and Spike protein negative; S pos, COVID-19 status positive and Spike-1 protein positive;" => can be omitted, since this is given in figure
- Fig.5: It would help to insert arrows to indicate macrophages and damaged pneumocytic cells in different colours in order to help non-experienced readers.
- Consistent writing eg line 265 "Sars-CoV2" => "SARS-CoV-2"; Fig.3: "COVID19" => "COVID-19"
- line 268: "machrophages" => "macrophages"
- Fig.6: maybe easier to read, if figure x-axis label/description as in Fig.3 => Negative / Positive; COVID-19 status
- Results section: lines 155-170 => Figure connections are missing.
- line 444: "someone comorbidities"? Whole sentence is unclear.
- table 1: What do the rows mean "COVID-19 (N.1)", "Control Cases (N.0)" etc for the age distinction?
Comments on the Quality of English Language
Some language issues are found, eg:
- line 174: "macropage" => "macrophage"
- line 176: "coorte" => "cohort"
- Consistent writing eg line 265 "Sars-CoV2" => "SARS-CoV-2"; Fig.3: "COVID19" => "COVID-19"
- line 268: "machrophages" => "macrophages"

Reviewer 2 Report
Thank you for sharing your manuscript on the role of TLR-4 in the context of macrophage imbalance among lethal COVID-19 disease and its association with Galectin-3. Here some some suggested edits and comments that could help to improve the article:
L431: Please specify in your manuscript which type of tissue was taken.
L475: Please add the version of SPSS used.
L453-457: Are these the controls mentioned in Table 1?
Table 1: Which controls? Please explain in your manuscript the 12 controls you have included. Also, the age classification <50yrs and >50yrs in relation to gender is not clear; consider re-structuring the table for more clarity.
L458-460: Consider moving or deleting this section. It does not fit here well and seems duplicated with L462-464.
General comment regarding "Conclusions" (L474-522): Please revise and shorten this section. Discussing your findings with published literature/other research conducted is usually done within the discussion section.
L1176: Not sure what "coorte" means.
General comment: Try to reduce the number of abbreviations; fewer will make your manuscript better readable. Also, common procedure is to introduce an abbreviation once only not several times, e.g. IHC L188 and L211.
See above.
Author Response
L431: Please specify in your manuscript which type of tissue was taken.
L475: Please add the version of SPSS used.
L453-457: Are these the controls mentioned in Table 1?
Table 1: Which controls? Please explain in your manuscript the 12 controls you have included. Also, the age classification <50yrs and >50yrs in relation to gender is not clear; consider re-structuring the table for more clarity.
L458-460: Consider moving or deleting this section. It does not fit here well and seems duplicated with L462-464.
General comment regarding "Conclusions" (L474-522): Please revise and shorten this section. Discussing your findings with published literature/other research conducted is usually done within the discussion section.
L1176: Not sure what "coorte" means.
General comment: Try to reduce the number of abbreviations; fewer will make your manuscript better readable. Also, common procedure is to introduce an abbreviation once only not several times, e.g. IHC L188 and L211.

Round 2
Reviewer 1 Report
The manuscript has improved to the first version. Please notify, that there still some issues to be resolved.
-
line 39: TLR-4 up-regulation is only seen in a subgroup of patients, which is even not the majority of patients.
- We have corrected sentences in the results paraghaph specifying that TLR-4 is up-rergulated in a subgroup of patients.
- => Please also change the text in the abstract (lines 41ff, 78ff).
- We have corrected sentences in the results paraghaph specifying that TLR-4 is up-rergulated in a subgroup of patients.
-
lines 465ff: Where are the results for all four viral RNA expressions?
- Correlation between PCR and ISH results have been previously published in a previous paper [Zito Marino, F.; De Cristofaro, T.; Varriale, M.; Zannini, G.; Ronchi, A.; La Mantia, E.; Campobasso, C.P.; De Micco, F.; Mascolo, P.; Municinò, M.; Municinò, E.; Vestini, F.; Pinto, O.; Moccia, M.; De Stefano, N.; Nappi, O.; Sementa, C.; Zotti, G.; Pianese, L.; Giordano, C.; Franco, R. Variable levels of spike and ORF1ab RNA in post-mortem lung samples of SARS-CoV-2-positive subjects: comparison between ISH and RT-PCR. Virchows Arch. 2022 Mar, 480(3), 597-607. doi: 10.1007/s00428-021-03262-8. Epub 2022 Feb 1. PMID: 35103846; PMCID: PMC8805427.] cited in the text with matched reference n33. Therefore in this study we presented the percentage of Spike1 positive cases as evaluated by ISH compared to mean values ± SEM of selected studied markers (TLR-4, CD68, CD163, Gal-3) adding a new table to the results section (table 3).
- Since RT-PCR is not used for the manuscript, 4.6 in the method section can be omitted.
- Correlation between PCR and ISH results have been previously published in a previous paper [Zito Marino, F.; De Cristofaro, T.; Varriale, M.; Zannini, G.; Ronchi, A.; La Mantia, E.; Campobasso, C.P.; De Micco, F.; Mascolo, P.; Municinò, M.; Municinò, E.; Vestini, F.; Pinto, O.; Moccia, M.; De Stefano, N.; Nappi, O.; Sementa, C.; Zotti, G.; Pianese, L.; Giordano, C.; Franco, R. Variable levels of spike and ORF1ab RNA in post-mortem lung samples of SARS-CoV-2-positive subjects: comparison between ISH and RT-PCR. Virchows Arch. 2022 Mar, 480(3), 597-607. doi: 10.1007/s00428-021-03262-8. Epub 2022 Feb 1. PMID: 35103846; PMCID: PMC8805427.] cited in the text with matched reference n33. Therefore in this study we presented the percentage of Spike1 positive cases as evaluated by ISH compared to mean values ± SEM of selected studied markers (TLR-4, CD68, CD163, Gal-3) adding a new table to the results section (table 3).
-
Please assign table numbers according to the order of appearance in the result section.
-
Fig. 3: Have you tried to split the positive COVID-19 group to spike negative/positive similar to Fig. 4? This might uncover significant differences.
-
Fig. 10: This figure seems to be dispensable, since Fig. 11 is a presentation of the same data in a more specific way.
-
Fig. 12: Comparison of the means GAL3 and CD163 is not suitable to detect association between these genes. However, correlation similar to Fig. 8 could help. Another option would be to skip this figure.
-
line 564: “Gal-3 antagonizes generation of a pro-inflammatory macrophage phenotype.” => Can be deleted, since similar to following sentence.
- Check again consistent writing of SARS-CoV-2, COVID-19, TLR4, GAL3, Mann-Whitney U-test, subgroup
- language issues, examples:
- line 199: "viral viral" => "viral"
- line 217: "cells infiltrates the lungs" => "cells infiltrate the lungs"
- lines 236: "In details," => "In detail"
- Fig. 1: "ialine" => "hyaline"?
Author Response
Response to Reviewer 1 Comments
Point 1: Line 39: TRL-4 up-regulation is only seen in a sugroup of patients, which is even not the maiority of patients
Response 1: We have specified that this study showed TRL-4 up-regulation in a subgroup of patients,
Point 2: We have corrected sentences in the results paragraph specifying that TRL-4 is up-regulated in a subgroup of patients.
Response 2: In the results paragraph and in the figure 4 we have corrected, specifying that TRL-4 is up-regulatded in a subroup of patients, throughout the text
Point 3: Please also change the text in the abstract (lines 41ff, 78ff)
Response 3: In the abstract we have changeD the text, specifyng that “this study showed TRL-4 up regulation in a subgroup of patients”
Point 4 Lines 465ff: where are the results for all four viral RNA expressions?
Response 4: We have eliminated sentences and paragraphs on PCR results
Point 5. Correlation between PCR and ISH results have been previously published in a previous paper [Zito Marino, F.; De Cristofaro, T.; Varriale, M.; Zannini, G.; Ronchi, A.; La Mantia, E.; Campobasso, C.P.; De Micco, F.; Mascolo, P.; Municinò, M.; Municinò, E.; Vestini, F.; Pinto, O.; Moccia, M.; De Stefano, N.; Nappi, O.; Sementa, C.; Zotti, G.; Pianese, L.; Giordano, C.; Franco, R. Variable levels of spike and ORF1ab RNA in post-mortem lung samples of SARS-CoV-2-positive subjects: comparison between ISH and RT-PCR. Virchows Arch. 2022 Mar, 480(3), 597-607. doi: 10.1007/s00428-021-03262-8. Epub 2022 Feb 1. PMID: 35103846; PMCID: PMC8805427.] cited in the text with matched reference n33. Therefore in this study we presented the percentage of Spike1 positive cases as evaluated by ISH compared to mean values ± SEM of selected studied markers (TLR-4, CD68, CD163, Gal-3) adding a new table to the results section (table 1).
Response 5: In the results paragraph we have specified that correlation between PCR and ISH results have been previously published in a previous paper [33]. Therefore in this study we presented the percentage of Spike1 positive cases as evaluated by ISH compared to mean values ± SEM of selected studied markers (TLR-4, CD68, CD163, Gal-3) adding a new table to the results section (Table 1, Figure 2).
Point 6. Since RT-PCR is not used for the manuscript, 4.6 in the method section can be omitted.
Response 6: In the materials and methods we have eliminated RT.PCR method.
Point 7. Please assign table numbers according to the order of appearance in the result section
Response 7: We have assegned table numbers according to the order of appearance in the result sections.
Point 8. Fig. 3: Have you tried to split the positive COVID-19 group to spike negative/positive similar to Fig. 4? This might uncover significant differences.
Response 8. •We have displayed in graphs TLR-4 espression according to Spike-1-positive and Spike-1 negative in a new version of figure 4, prefering to keep figure 3 unchanged, displaying very well histological aspects.
Point 9. Fig. 10: This figure seems to be dispensable, since Fig. 11 is a presentation of the same data in a more specific way.
Response 9. •We removed figure 10, and we left figure 11 (now called figure 10 in the new version of the paper)
Point 10. Fig. 12: Comparison of the means GAL3 and CD163 is not suitable to detect association between these genes. However , correlation similar to Fig. 8 could help. Another option would be to skip this figure.
Response 10. We have changed wrong figure 12 with a graph showing the linear correlation between GAL3 and CD163 by the SPEARMAN’S correlation test. New correct past figure 12 is called figure figure 11 in accordance with the new numbering of the figures in the text. Comments on this new figure have been introduced in the text.
Point 11. Gal-3 antagonizes generation of a pro-inflammatory macrophage phenotype.” => Can be deleted, since similar to following sentence.
Response 11. We have deleted the sentence “Gal-3 antagonizes generation of pro-inflammatory macrophage phenotype”
Comments on the Quality of English Language
- Check again consistent writing of SARS-CoV-2, COVID-19, TLR4, GAL3, Mann-Whitney U-test, subgroup We have corrected
- language issues, examples:
- line 199: "viral viral" => "viral". We have corrected line 199
- line 217: "cells infiltrates the lungs" => "cells infiltrate the lungs" We have corrected line 217
- lines 236: "In details," => "In detail" We have corrected line 236
- Fig. 1: "ialine" => "hyaline"? We have corrected figure 1

Reviewer 2 Report
Thank you for addressing most of my comments in the revised manuscript. I leave it to the editorial board of IJMS how to proceed with the issue of abbreviations, which should be kept to a necessary minimum and introduced once only when used the first time in the manuscript.
Please see above.
Author Response
There is not reviewe comment.